# Quantitative microscopy of functional HIV post-entry complexes reveals association of replication with the viral capsid

Ke Peng[1], Walter Muranyi[1], Bärbel Glass[1], Vibor Laketa[1,2], Stephen R Yant[3], Luong Tsai[3], Tomas Cihlar[3], Barbara Müller[1], Hans-Georg Kräusslich[1,2]*

[1]Department of Infectious Diseases, Virology, University of Heidelberg, Heidelberg, Germany; [2]German Center for Infection Research, partner site Heidelberg, Germany; [3]Gilead Sciences Inc., Foster City, United States

**Abstract** The steps from HIV-1 cytoplasmic entry until integration of the reverse transcribed genome are currently enigmatic. They occur in ill-defined reverse-transcription- and pre-integration-complexes (RTC, PIC) with various host and viral proteins implicated. In this study, we report quantitative detection of functional RTC/PIC by labeling nascent DNA combined with detection of viral integrase. We show that the viral CA (capsid) protein remains associated with cytoplasmic RTC/PIC but is lost on nuclear PIC in a HeLa-derived cell line. In contrast, nuclear PIC were almost always CA-positive in primary human macrophages, indicating nuclear import of capsids or capsid-like structures. We further show that the CA-targeted inhibitor PF74 exhibits a bimodal mechanism, blocking RTC/PIC association with the host factor CPSF6 and nuclear entry at low, and abrogating reverse transcription at high concentrations. The newly developed system is ideally suited for studying retroviral post-entry events and the roles of host factors including DNA sensors and signaling molecules.

*For correspondence: hans-georg.kraeusslich@med.uni-heidelberg.de

**Competing interests:** The authors declare that no competing interests exist.

## Introduction

The early phase of human immunodeficiency virus (HIV-1) replication involves reverse transcription of the viral RNA genome in the cytoplasm followed by nuclear entry of the linear double-stranded viral cDNA and integration into a host cell chromosome. Virus entry by membrane fusion releases the viral capsid containing genomic RNA and viral structural and replication proteins into the cytoplasm. While much is known about the biochemistry and inhibition of reverse transcription (*Hu and Hughes, 2012*), the processes of uncoating, genome replication, and nuclear trafficking are currently not well understood in infected cells. Several viral and host factors have been implicated in these steps, but their precise contributions are largely unknown. Different models have been proposed for genome uncoating, which is defined as the loss of the viral capsid structure that consists of homomultimers of CA (capsid) proteins: (i) immediate dissociation after fusion followed by reverse transcription within the remaining nucleoprotein complex; (ii) gradual dissociation during reverse transcription regulated by sequential contact with host factors; (iii) reverse transcription in an intact or largely intact capsid structure that only dissociates at the nuclear pore complex (reviewed in *Arhel (2010)*). Several recent studies indicated that mutations in CA as well as small molecule compounds targeting CA can affect reverse transcription and nuclear import (reviewed in *Matreyek and Engelman (2013)*), disfavoring the model of immediate capsid dissociation. Furthermore, various host factors interacting with CA have been implicated in the early phase of HIV-1 replication, again arguing for a functional contribution of CA at this stage (reviewed in *Matreyek and Engelman (2013)*; *Ambrose and Aiken (2014)*; *Hilditch and Towers (2014)*). Reverse transcription in the shielded capsid environment could prevent

**eLife digest** Major advances in the treatment of HIV have been made possible by carefully studying the virus and its interaction with the host cell. The virus consists of two strands of RNA—representing the genetic information of the virus—contained in a protein coat called capsid. Scientists have learned that the virus' RNA is used to create viral DNA in the cytoplasm of an infected cell, in a process called reverse transcription. This viral DNA then enters the cell's nucleus and becomes incorporated into the cell's DNA, and the cell unwittingly begins to help the virus reproduce.

It is less clear what happens to the capsid after the virus enters a cell. Some researchers have suggested that it is lost shortly after entry or during reverse transcription. However, some recent studies have found that damaging the capsid hampers reverse transcription and significantly impairs the entry of viral DNA into the cell's nucleus. This suggests that the capsid might continue to protect the viral genome when the RNA is converted into DNA.

To learn more about what happens during reverse transcription and when the viral DNA enters the nucleus, it is important to watch individual events as they occur. Until recently, it had been hard to do this without changing the DNA or RNA in ways that might affect their properties. Recently, a technique called click-labeling has been developed that can add a fluorescent label to DNA or RNA without potentially damaging this genetic material. This label allows the movement of the DNA or RNA to be followed when the cell is viewed under a microscope.

Peng et al. used this new technique to watch reverse transcription, how viral DNA enters the cell nucleus and what happens to the capsid when HIV invades different kinds of cells. When the virus entered a type of cell often used in laboratory research called HeLa cells, the capsid protected the viral genetic material when it was in the cell's cytoplasm but disappeared before or shortly after the viral DNA entered the cell's nucleus. However, HeLa cells are not natural targets of HIV; when Peng et al. looked at the behavior of the capsid in the immune cells that the virus normally invades, the capsid was present in both the cytoplasm and the nucleus of these cells.

Peng et al. also observed what happens in HIV-infected cells treated with a chemical called PF74 that interferes with the capsid. This revealed that low concentrations of PF74 make it hard for the viral DNA to enter the nucleus, probably by blocking the interaction of the capsid with a protein from the host cell. At high concentrations, the drug prevented reverse transcription.

The approach used by Peng et al. allows direct visualization of how HIV replicates and how this DNA is imported into the nucleus of cells naturally targeted by the virus. This will aid our understanding of how the virus selects where in the host genome it should insert its DNA, which is important for establishing a permanent infection in the cell.

HIV-1 from recognition by cytoplasmic DNA sensors with subsequent capsid uncoating at the nuclear pore considered to be required for nuclear entry (*Matreyek and Engelman, 2013*). Productive retroviral reverse transcription complexes (RTC) and pre-integration (PIC) complexes have been defined biochemically as high molecular weight complexes that after isolation from cells support endogenous reverse transcription (*Fassati and Goff, 1999*) and integrate viral cDNA into heterologous target DNA (*Brown et al., 1987*), respectively. However, the lack of methods for robust and reliable detection so far prevented direct determination of the association of functional RTC/PIC with specific viral or cellular proteins at different post-entry stages.

The majority of incoming HIV-1 particles fails to transform into productive replication complexes in most commonly used cell lines (*Thomas et al., 2007b*). Accordingly, detection of viral constituents alone does not allow identifying functional structures over the large background of non-functional complexes. Furthermore, HIV-1 replication complexes are inherently transient and unstable complicating biochemical analysis, and different constituents were reported in different biochemical studies (*Farnet and Haseltine, 1991*; *Bukrinsky et al., 1993*; *Miller et al., 1997*; *Fassati and Goff, 2001*). Identifying functional HIV-1 RTC/PIC by imaging techniques would allow in situ investigation but requires the detection of nascent HIV-1 cDNA in association with viral constituents. A pioneering study applied microinjection of fluorescent nucleotides and subsequent HIV-1 infection, detecting microtubule association of a replication complex by correlative light and negative stain electron microscopy

(*McDonald et al., 2002*). However, no further information on the RTC could be derived in this study, and the method did not permit reliable identification of larger numbers of RTC/PIC. Accordingly, no follow-up studies applying this technique have been published.

Specific detection of HIV-1 cDNA could be achieved by fluorescence in situ hybridization, and this method has been used to visualize integrated HIV-1 cDNA (*Lusic et al., 2013*). Processing steps during FISH interfere with immunostaining (*Solovei and Cremer, 2010*), however, making detection of productive HIV-1 RTC/PIC difficult. Alternatively, viral cDNA may be detected by non-specific labeling. Classically, DNA synthesis has been visualized by bromo-deoxyuridine incorporation (*Gratzner et al., 1975*), which again requires denaturation. More recently, labeling nascent DNA with a nucleoside analogue carrying an alkyne group for bio-orthogonal 'click-labeling' with a fluorophore carrying an azide group has been introduced (*Salic and Mitchison, 2008*). Click-labeling can be directly performed without denaturation or other processing steps. This novel approach has recently been used successfully for the investigation of SV40 disassembly using pre-labeled particles (*Kuksin and Norkin, 2012*) and for the detection of replication factories of several DNA viruses (*Strang et al., 2012*; *Wang et al., 2013*). Many viral genomes are rapidly produced in a confined space in these cases, however. In contrast, HIV-1 reverse transcription produces only a single cDNA copy from the ~10 kb RNA genome, thus requiring a high sensitivity of detection.

Here, we describe a robust strategy for detecting HIV-1 RTC/PIC using click-labeling of 5-ethynyl-2'-deoxyuridine (EdU) incorporated into nascent viral DNA. Co-localization of a cytoplasmic EdU signal with GFP-tagged HIV-1 integrase (IN) could be used to reliably detect HIV-1 RTC/PIC in a reporter cell line and primary human macrophages. Specific immunostaining revealed the presence of the HIV-1 CA and NC (nucleocapsid) proteins on almost all cytoplasmic RTC/PIC, while CA was absent from nuclear PIC in a HeLa-derived reporter cell line. In contrast, almost all nuclear PIC were positive for CA in primary human macrophages, while CA detection on cytoplasmic RTC/PIC was more variable in this case. EdU labeling further allowed determination of a dual mode of action for the recently described HIV-1 inhibitor PF-3450074 (PF74; [*Shi et al., 2011*]) targeting the CPSF6 binding interface of CA. The new method for identification of functional retroviral replication complexes will be widely applicable for studying the RTC/PIC pathway and the roles of host factors, DNA sensors and signaling molecules, as well as association with the nuclear import machinery.

## Results

To identify and characterize productive HIV-1 RTC/PIC during the early post-entry stages of infection, we made use of the fact that synthesis of the genomic HIV-1 cDNA occurs in the cytoplasm, that is spatially separated from nuclear DNA replication of the host cell. When restricting analysis to extranuclear regions, HIV-1 reverse transcription thus only needs to be distinguished from mitochondrial DNA synthesis. Accordingly, fluorescent labeling of the nascent HIV-1 cDNA using EdU incorporation and click-labeling should allow unequivocal detection of individual productive RTC/PIC when combined with labeling for a component of the viral replication machinery and a mitochondrial marker. To label HIV-1 specific complexes, we made use of a previously described IN.eGFP fusion protein that remains associated with incoming subviral particles throughout early replication and nuclear import (*Albanese et al., 2008*).

### Identification of HIV-1 RTC/PIC

Infectious HIV-1 particles carrying IN.eGFP were produced by co-transfection of HEK 293T cells with the respective plasmids. The majority of virions contained microscopically detectable IN.eGFP (*Figure 1—figure supplement 1*). IN.eGFP was incorporated in similar amounts as virus-encoded IN (*Figure 1—figure supplement 2*), and incorporation of IN.eGFP reduced infectivity only weakly (*Figure 1—figure supplement 3A*). Detection of (potentially weak) cytoplasmic DNA signals may be obscured by very strong labeling of replicating nuclear DNA. Indeed, initial EdU labeling experiments revealed bright nuclear fluorescence that prevented analysis of fluorescent signals in the extranuclear region of the same and neighboring cells (*Figure 1—figure supplement 4*). This problem was overcome by infecting cells in the presence of the DNA polymerase α/δ inhibitor aphidicolin (APC). Cell cycle arrest by APC is compatible with HIV-1 replication, and APC caused only a modest reduction of infectivity (*Figure 1—figure supplement 3B*) since HIV-1 does not require nuclear envelope breakdown for genome integration (*Suzuki and Craigie, 2007*). Infecting cells at APC concentrations that did not exhibit toxic effects during the incubation period led to a strong reduction of the nuclear EdU

signal, now allowing detection of cytoplasmic EdU signals (*Figure 1—figure supplement 4*). The remaining nuclear EdU signal reflects residual nuclear DNA replication at this APC concentration.

HeLa-based reporter cells (TZM-bl) were pre-incubated with HIV-1 (IN.eGFP) at a high multiplicity of infection (m.o.i. = 25) for 30 min at 16°C to synchronize the infection. Cells were subsequently shifted to 37°C and infection was performed in the presence of EdU for 4 hr, followed by fixation and click-labeling using Alexa Fluor 647 (*Figure 1A*). Immunostaining of cytochrome C (CC) was carried out in addition to mark mitochondria. Spinning disc confocal microscopy (SDCM) revealed large numbers of intracellular eGFP-labeled particles, lacking EdU or CC labeling (*Figure 1A*, large panels, green signal). Most of these objects probably represent virions taken up by non-specific heparan sulfate-mediated endocytosis known to occur efficiently in HeLa-derived cells (*Lampe et al., 2007*). Importantly, some eGFP-labeled particles co-localized with distinct, punctate EdU signals in the extra-nuclear region without overlapping mitochondria labeling (*Figure 1A*, insets and enlargements i–iii; compare *Figure 1—figure supplement 5* for staining of mitochondrial DNA synthesis). Co-localization was defined as objects in which Alexa Fluor 647- and eGFP signals overlapped in the major part of the pixel area in several adjacent frames of the confocal z-stack.

To validate the co-localization of labeled IN with newly synthesized EdU-labeled DNA in subviral complexes, we performed super-resolution microscopy. Cells were infected with virions carrying a photo-switchable IN.mEos3.2 fusion protein (*Zhang et al., 2012*), suitable for detection with photoactivated localization microscopy (PALM) (*Betzig et al., 2006*) and click-labeled with EdU. Dual-color PALM/dSTORM (*Heilemann et al., 2008*) yielded a lateral resolution of 46 nm and 32 nm for mEos3.2 and Alexa Fluor 647, respectively. In order to ensure high resolution in all three dimensions, samples were imaged in total internal reflection mode. This method limits detection to the close proximity (~200 nm) of the ventral plasma membrane, where only a small proportion of all RTC/PIC present in the cell is expected to be located. Nevertheless, in two independent experiments we detected 13 clear EdU/IN co-localizing objects that displayed partial overlap of the two signals at super-resolution precision (*Figure 1B,i,ii*). Such objects were also observed deeper in the cytoplasm when employing PALM/dSTORM in epifluorescence mode (*Figure 1B,iii,iv*).

To determine whether IN.eGFP/EdU co-localization is suitable for identifying productive HIV-1 RTC/PIC, we performed experiments in the presence of the HIV-1 reverse transcriptase inhibitor efavirenz (EFV). Approximately 68% of cells infected in the absence of EFV displayed IN.eGFP/EdU-co-localizing objects at 4.5 hr p.i., with up to 30 objects in individual cells and an average of five co-localizing objects per cell (*Figure 1C*). Control experiments verified that individual viral cDNA molecules and also partially reverse transcribed intermediates should be easily detectable in our setup (*Figure 1—figure supplement 6*). The wide variation in the number of co-localizing eGFP- and EdU-positive objects per cell (*Figure 1—figure supplement 7*), observed even within one experiment, reflects the stochasticity of the infection process.

Infection in the presence of EFV did not affect the uptake of IN.eGFP particles but dramatically reduced the number of IN.eGFP/EdU co-localizing objects to 0.2 per cell (*Figure 1C*). We therefore defined IN.eGFP/EdU co-localization as the signature of productive HIV-1 replication complexes. Since the transition from RTC to PIC is defined biochemically and our imaging analyses do not allow discriminating between these complexes in the cytoplasm, we designated IN.eGFP/EdU-positive subviral particles in the cytoplasm of infected cells as RTC/PIC.

We subsequently determined the time course of RTC/PIC formation. RTC/PIC were detected in almost half of the cells at 2 hr after infection, with the majority of positive cells containing a single object (*Figure 2*). The proportion of RTC/PIC containing cells, as well as the average number of complexes per cell, increased significantly when the infection period was extended to 3 hr. Infection for 4 hr led to a further increase, while no significant additional RTC/PIC accumulation was detected at later time points (*Figure 2*). Based on this result, an incubation time of 4–5 hr was chosen for all further experiments in HeLa-derived cells.

## Association of RTC/PIC with HIV-1 structural proteins

We subsequently analyzed whether the main HIV-1 structural proteins MA (matrix), CA, or NC remain associated with productive RTC/PIC in the early phase of HIV-1 replication. Cells were infected with HIV-1 (IN.eGFP) for 4 hr in the presence of EdU followed by click-labeling and immunostaining (*Figure 3A*). Numerous intracellular particles exhibiting co-localization of IN.eGFP with the viral structural proteins, but lacking an EdU signal, were observed in all cases (*Figure 3A*). Analysis

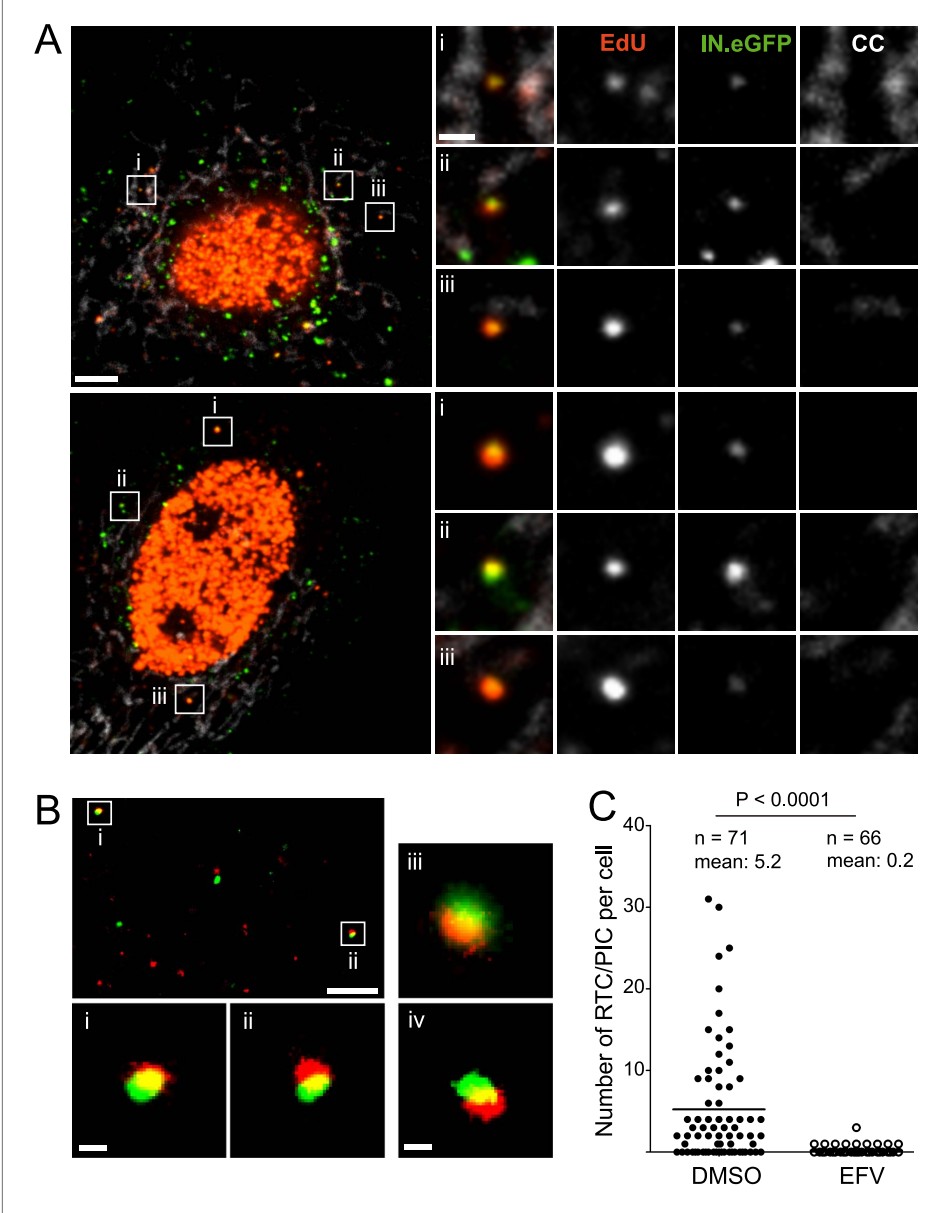

**Figure 1**. HIV-1 RTC/PIC detection in TZM-bl cells. (**A**) Identification of candidate RTC/PIC based on co-localization of EdU with IN.eGFP. TZM-bl cells were infected with HIV-1 (IN.eGFP) for 4 hr in the presence of APC and EdU, followed by fixation and click-labeling. Mitochondria were detected by cytochrome C immunostaining (CC; white). z-stacks covering the whole-cell volume were acquired by SDCM and analyzed for distinct co-localizing EdU-AlexaFluor647 (red) and IN.eGFP (green) signals. Left panels show z-sections through representative cells. Boxed regions (**i–iii**) mark three co-localizing objects from each cell displayed as enlargements on the right. Images from the optimal focal plane of each object are shown in the enlargements. Scale bars: 5 μm (overviews), 1 μm (enlargements). (**B**) Super-resolution microscopy confirming co-localization between EdU-AlexaFluor647 (red) and IN.mEos3.2 (green). TZM-bl cells were infected with HIV-1 (IN.mEos3.2) for 4 hr before click-labeling and analyzed by dual-color PALM/dSTORM. The overview shows an image recorded in TIRF mode; boxed regions **i–ii** are enlarged below. Enlargements **iii–iv** show additional examples of co-localization recorded by PALM/dSTORM in epifluorescence mode. Scale bars: 1 μm (overview), 100 nm (enlargements). (**C**) Inhibition of reverse transcriptase prevents RTC/PIC formation. TZM-bl cells were infected with HIV-1 (IN.eGFP) for 4.5 hr in the presence of DMSO or 5 μM EFV, respectively, followed by click-labeling and immunostaining as in (**A**). z-stacks covering the whole-cell volume were acquired for randomly selected cells and the number of RTC/PIC per cell was determined. The graph shows pooled results from three independent experiments. Statistical significance was assessed using the

*Figure 1. Continued on next page*

*Figure 1. Continued*

Mann–Whitney test. See *Figure 1—figure supplement 1–3* for characterization of labeled virus particles and effect of APC on infectivity, *Figure 1—figure supplement 4* for reduction of nuclear EdU signal with APC treatment, *Figure 1—figure supplement 5* for detection of mitochondrial DNA in the cytoplasm, *Figure 1—figure supplement 6* for analysis of detection sensitivity, and *Figure 1—figure supplement 7* for summary of number of RTC/PIC detected per cell.

The following figure supplements are available for figure 1:

**Figure supplement 1**. Analysis of labeled virus particles by immunofluorescence.

**Figure supplement 2**. Analysis of labeled virus particles by immunoblot.

**Figure supplement 3**. Quantification of relative infectivity of labeled virions.

**Figure supplement 4**. Reduction of nuclear EdU signal by APC treatment.

**Figure supplement 5**. Detection of mitochondrial DNA synthesis in the cytoplasm.

**Figure supplement 6**. Sensitivity of EdU-Alexa Fluor647 detection.

**Figure supplement 7**. Numbers of cytoplasmic RTC/PIC detected per cell.

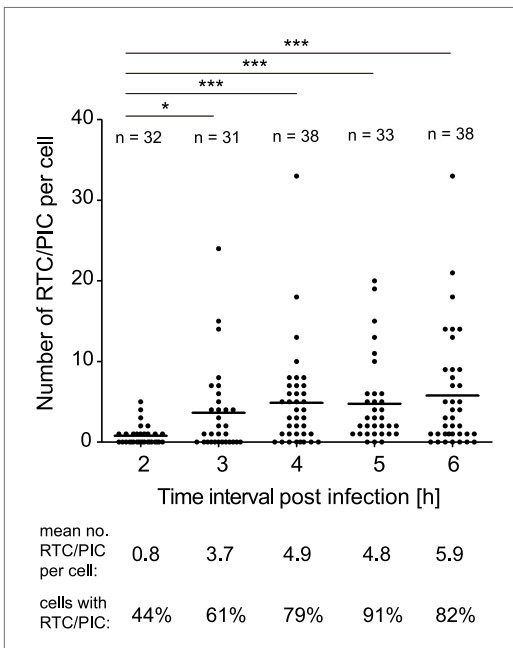

**Figure 2**. Time course of RTC/PIC detection. TZM-bl cells were infected with HIV-1 (IN.eGFP) as in *Figure 1*. Cells were fixed at the indicated time points after infection and analyzed as in *Figure 1C*. Numbers of RTC/PIC detected per cell were counted for randomly selected cells in each incubation interval. The graph shows pooled results from two independent experiments. Mean values of RTC/PIC per cell are indicated by black lines. Statistical significance was assessed using the Mann–Whitney test.

of >100 RTC/PIC per condition revealed co-localization with easily detectable NC and CA signals in ~95% of both cases (*Figure 3B*). In contrast, MA co-localization was only detected in 6% of the RTC/PIC analyzed (*Figure 3B*), while the MA signal was easily detectable on most intracellular IN.eGFP-positive objects lacking the EdU signal (*Figure 3A*). Intensity of CA and NC staining on RTC/PIC was variable. Quantitative comparison of the CA signals on >100 RTC/PIC with an equal number of EdU-negative, but IN.eGFP and CA-positive objects (many of which are likely to correspond to complete particles in endosomes) revealed a slightly higher average intensity on RTC/PIC (*Figure 3C*). This may be due to different exposure of conformational epitopes that can be recognized by our polyclonal antiserum. IN.eGFP signals were almost equal in both cases.

The observation that almost all cytoplasmic RTC/PIC co-localized with a clearly detectable CA signal indicated that reverse transcription occurs within the incoming capsid or a CA-containing structure in this cell type. To validate this co-localization, we performed three-color PALM/dSTORM super-resolution microscopy detecting (i) IN.mEos3.2, (ii) EdU-Alexa Fluor 647, and (iii) CA, using antibodies labeled with Alexa Fluor 532. This approach yielded a localization precision of 20 nm for the Alexa Fluor 532 label. Co-localization of the CA-specific signal with RTC/PIC could indeed be validated at super-resolution (*Figure 4A*), with 16 co-localizing objects observed

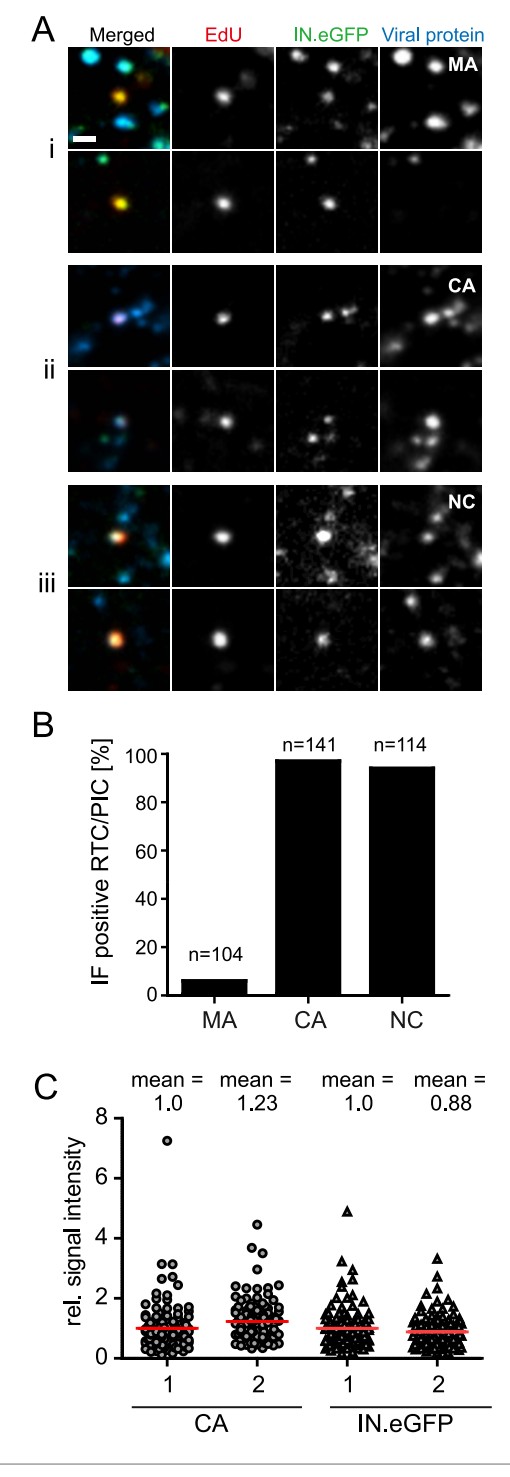

**Figure 3**. Association of HIV-1 Gag derived proteins with RTC/PIC. (**A**) Immunofluorescence analysis. TZM-bl cells were infected with HIV-1 (IN.eGFP) as in *Figure 1*. At 4 hr p.i. samples were fixed, click-labeled, and immunostained using antisera against HIV-1 MA (**i**), CA (**ii**), or NC (**iii**), respectively. z-stacks covering the whole-cell volume were acquired by SDCM, RTC/PIC

in eight different cells. We measured an average diameter of ~155 nm for the CA signal and ~133 nm for the mEos3.2 signal with modest variation and no clear outliers (*Figure 4—figure supplement 1*).

The subcellular localization of HIV-1 reverse transcription is not fully established due to a lack of experimental approaches detecting productive RTC. Our analysis revealed IN/EdU-positive objects in the cell periphery and deeper in the cytoplasm (*Figure 1A,B*). To determine whether RTC/PIC in the cell periphery and in the perinuclear region exhibit differences in CA association, we performed SDCM using an antibody against nucleoporins to mark the nuclear envelope. RTC/PICs in the periphery (*Figure 4B,i*), close to the nucleus (*Figure 4B,ii*) or co-localizing with nuclear pore staining (*Figure 4B,iii*) all contained CA. Nuclear HIV-1 specific EdU-positive particles (corresponding to PIC) could not be unambiguously identified in most cells, since EdU signals from residual chromosomal DNA replication prevented their detection, even under APC treatment (*Figure 1—figure supplement 4*). Occasional cells did not show EdU labeling of chromosomal DNA, however (presumably due to the absence of cellular DNA synthesis during the incubation period). IN.eGFP/EdU-positive nuclear PIC could be detected in these cells, and the nuclear PIC exhibited no detectable CA staining in almost all (23/24) cases (*Figure 4C*). Thus, cytoplasmic RTC/PIC are clearly associated with CA independent of their subcellular localization, while nuclear PIC have lost most or all of their CA content, at least in this HeLa-derived cell line.

## Bimodal inhibition of HIV-1 replication by the assembly inhibitor PF74

The small molecule inhibitor PF74 has been reported to abrogate HIV-1 replication prior to the stage of cDNA synthesis (*Blair et al., 2010*; *Shi et al., 2011*). PF74 binds a groove in the N-terminal domain of HIV-1 CA and has been proposed to induce premature uncoating with consequent loss of reverse transcription (*Shi et al., 2011*; *Matreyek et al., 2013*). Furthermore, PF74 has been reported to interfere with association of the RTC with the host proteins CPSF6 and Nup153 (*Matreyek et al., 2013*). We confirmed that PF74 completely blocks HIV-1 infection at a concentration of 10 μM (*Figure 5A*) as previously shown (*Shi et al., 2011*). Inhibition affected a stage preceding cDNA synthesis: detection of RTC/PIC was dramatically reduced in cells infected for 4.5 hr in the presence of 10 μM PF74

*Figure 3. Continued*

were identified as in *Figure 1* and co-localization with each Gag derived protein was analyzed. The panel shows two representative RTC/PIC per antiserum used. Scale bar: 1 µm. (**B**) Quantitative analysis of co-localization with viral proteins. For each staining condition, n > 100 individual randomly selected RTC/PIC, identified as in (**A**), were analyzed for detection of the respective Gag derived protein. 6%, 97%, and 94% of RTC/PIC were positive for MA, CA, or NC, respectively. (**C**) Quantification of relative signal intensity. A total of 114 intracellular CA-IN.eGFP objects not containing detectable EdU signal (1) and 114 RTC/PIC (2), respectively, were identified in cells from three independent experiments. Signal intensity of CA (circles) and IN.eGFP (triangles) was calculated for each object as described in 'Materials and methods'. Within each experiment, values were normalized to the mean intensity of CA or IN.eGFP signal, respectively, that was obtained for the CA-IN.eGFP objects lacking EdU. Each data point in the graph represents a single object; mean intensity values are indicated by red lines.

per cell on average, respectively; *Figure 5B*). Furthermore, infection in the presence of 2 µM PF74 did not affect association of RTC/PIC with CA: 98% and 97% of RTC/PIC exhibited a positive CA signal when cells were infected in the presence or absence of 2 µM PF74, respectively (*Figure 5B*).

The result that reverse transcription was unaffected by ~2 µM PF74 was confirmed by quantitative PCR analysis. No significant difference was observed for late reverse transcription products when cells were infected with HIV-1 in the presence or absence of 2.7 µM PF74, while these products were lost upon infection in the presence of EFV or at a higher PF74 concentration, respectively (*Figure 5C*). A different result was observed when HIV-1 specific 2-LTR circles, indicative of nuclear import of viral DNA, were quantified. Infection in the presence of 2.7 µM PF74 almost completely abolished the formation of 2-LTR circles despite normal synthesis of late reverse transcription products, indicating a specific defect in nuclear import of viral cDNA. As expected, EFV or 8.1 µM PF74 abolished 2-LTR circles as well, while incubation in the presence of the IN inhibitor elvitegravir (EVG), which blocks chromosomal integration of the HIV-1 cDNA without affecting reverse transcription or nuclear import (*Shimura et al., 2008*), yielded a strong increase of 2-LTR circles (*Figure 5C*). Taken together these results indicate a dose-dependent bimodal effect of PF74 on early HIV-1 infection, targeting reverse transcription and PIC nuclear import, respectively.

The nuclear import defect at low concentrations of PF74 may be caused by binding of this compound to a reactive groove on the viral capsid, leading to competitive inhibition of capsid interaction with the host proteins Nup153 and CPSF6 (*Matreyek et al., 2013*; *Price et al., 2014*). Both of these proteins have been implicated in HIV-1 PIC nuclear import (*Matreyek and Engelman, 2013*). We therefore investigated the association of RTC/PIC with CPSF6 in the presence and absence of PF74. Immunostaining of CPSF6 revealed an almost exclusively nuclear localization of this protein with very weak cytoplasmic staining in TZM-bl cells (*Figure 6—figure supplement 1A*), consistent with previously published reports (*De Iaco et al., 2013*; *Fricke et al., 2013*). Nevertheless, a clear CPSF6 signal co-localizing with the RTC/PIC was detected in 22% of all cases in TZM-bl cells (*Figure 6A*; 92 RTC/PIC analyzed). Given the very weak cytoplasmic CPSF6 signal, we considered the possibility that this relatively low number of co-localizing structures may be due to insufficient sensitivity of our detection system. To overcome this obstacle, we made use of a HeLa-derived cell line with a stable knock-down of transportin-3 (TNPO3) (*Thys et al., 2011*). This protein functions as a nuclear import factor for CPSF6, and TNPO3 knock-down has been shown to lead to cytoplasmic accumulation of CPSF6 (*De Iaco et al., 2013*). Accordingly, an increased cytoplasmic CPSF6 signal was detected in the TNPO3 knock-down cell line but not in a control cell line expressing a scrambled shRNA (*Figure 6—figure supplement 1B,C*). Consistent with our hypothesis, we observed that 87% of all RTC/PIC were positive

(*Figure 5B*), and this result was confirmed by quantitative determination of HIV-1 specific cDNA using PCR (*Figure 5C*). The observation that the remaining RTC/PIC observed at this PF74 concentration were significantly less frequently positive for CA (65% vs 97% in the control sample; p < 0.01) is consistent with the proposed effect of PF74 on capsid uncoating.

Titration of PF74 had revealed an $EC_{50}$ for inhibition of HIV-1 infection of ~0.5 µM in MT-2 cells (*Blair et al., 2010*). In HeLa-derived cell lines, 80–90% reduction of infectivity was observed at a concentration of 1 or 2 µM PF74, respectively (*Shi et al., 2011*). 2 µM PF74 reduced HIV-1 infectivity >95% in our experimental system, even at the high multiplicity of infection used. Unexpectedly, however, RTC/PIC formation was completely unaffected at this PF74 concentration, despite almost complete loss of infectivity. EdU-positive RTC/PIC were readily detected in cells infected with HIV-1 for 4.5 hr in the presence of 2 µM PF74 with no apparent difference compared to control infections (7.7 vs 6.8 RTC/PIC

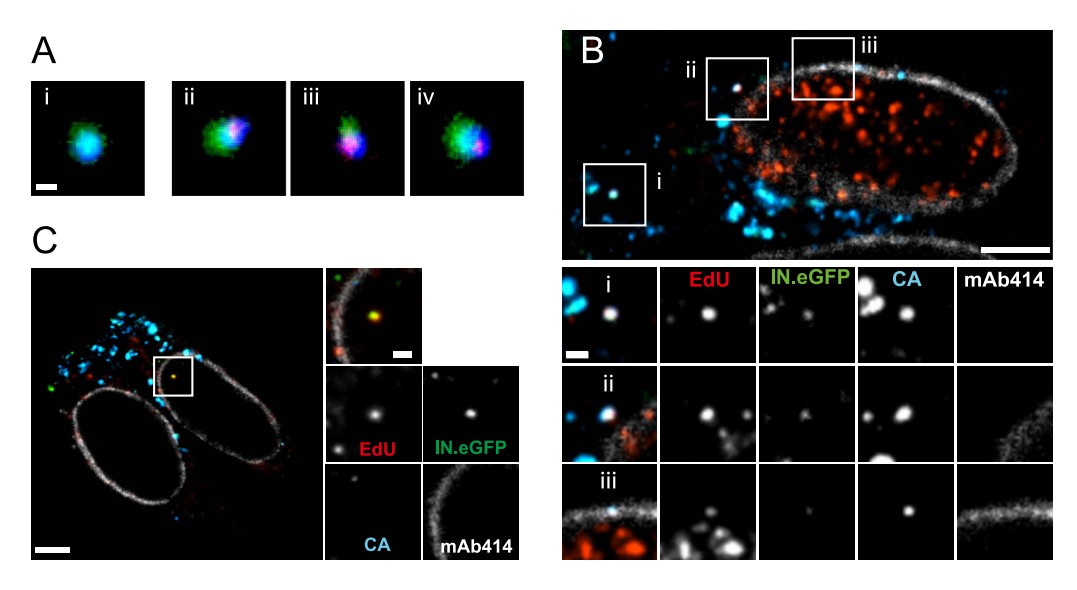

**Figure 4**. Co-localization of HIV-1 CA with RTC/PIC at different intracellular localizations. (**A**) Three-color PALM/dSTORM analysis of RTC/PIC. TZM-bl cells were infected with HIV-1 (IN.mEos3.2) as in *Figure 1*. At 4 hr p.i., cells were fixed, click-labeled, and immunostained using antiserum against HIV-1 CA. Three-color PALM/dSTORM images were acquired in TIRF mode as described in 'Materials and methods'. The panel shows an example of an EdU-negative IN.mEos3.2/CA co-localizing object that possibly represents an intracellular virion (**i**), as well as three examples of triple-labeled complexes classified as RTC/PIC (**ii–iv**). Scale bar: 100 nm. (**B**) RTC/PIC detected in the cytoplasmic and perinuclear regions. TZM-bl cells were infected with HIV-1 (IN.eGFP) as in *Figure 1*. At 5 hr p.i., samples were fixed, click-labeled, and immunostained using an antibody against nuclear pore proteins (white) and antiserum against HIV-1 CA (blue). The top panel shows a *z*-section through an individual cell displaying RTC/PIC in the peripheral cytoplasmic region (**i**), close to the nucleus (**ii**), and overlapping or in close proximity of the nuclear membrane (**iii**). The lower panels show enlargements of the insets with images selected from the optimal focal plane of each RTC/PIC. Scale bars: 5 μm (overview), 1 μm (enlargements). (**C**) PIC detected inside the nucleus. The left panel shows a *z*-section through two individual cells, selected for very low levels of nuclear EdU labeling. Immunostaining was performed as in (**B**). The boxed area shows an IN.eGFP/EdU co-localizing object within the nucleus. Enlargements of this object in the individual channels are shown on the right. Scale bars: 5 μm (overview), 1 μm (enlargements). See *Figure 4—figure supplement 1* for analysis of cluster sizes of RTC/PIC detected in three-color PALM/dSTORM analysis.

The following figure supplement is available for figure 4:

**Figure supplement 1**. Size distribution of RTC/PIC associated CA and IN.mEos clusters derived from PALM/dSTORM analyses.

for CPSF6 upon infection of the TNPO3 knock-down cell line (*Figure 6B*; 87 RTC/PIC analyzed). Treatment of TNPO3 knock-down cells with 2 μM PF74 during infection strongly reduced the level of CPSF6 association with RTC/PIC to 18% (*Figure 6C*; 74 RTC/PIC analyzed). These results provide direct evidence for the association of cytoplasmic CPSF6 with the incoming viral capsid and for competitive inhibition of this interaction by the small molecule inhibitor PF74.

## HIV-1 RTC/PIC formation in primary human macrophages

All imaging experiments described so far were performed in a HeLa-derived cell line, which is well suited for microscopic analyses and allows robust detection of RTC/PIC. These cells do not represent natural target cells of HIV-1, however. We therefore extended our analysis to primary human monocyte-derived macrophages (MDM). MDM were chosen because they are natural targets of HIV-1 infection with a morphology well suited for microscopy, and they are post-mitotic cells lacking nuclear DNA synthesis. The experiments with MDM were performed using HIV-1 carrying Env proteins with a tropism for the CCR5 co-receptor. IN.eGFP/EdU-positive RTC/PIC could be detected in HIV-1 infected MDM (*Figure 7*), albeit at a much lower frequency compared to TZM-bl cells. HIV-1 uptake into

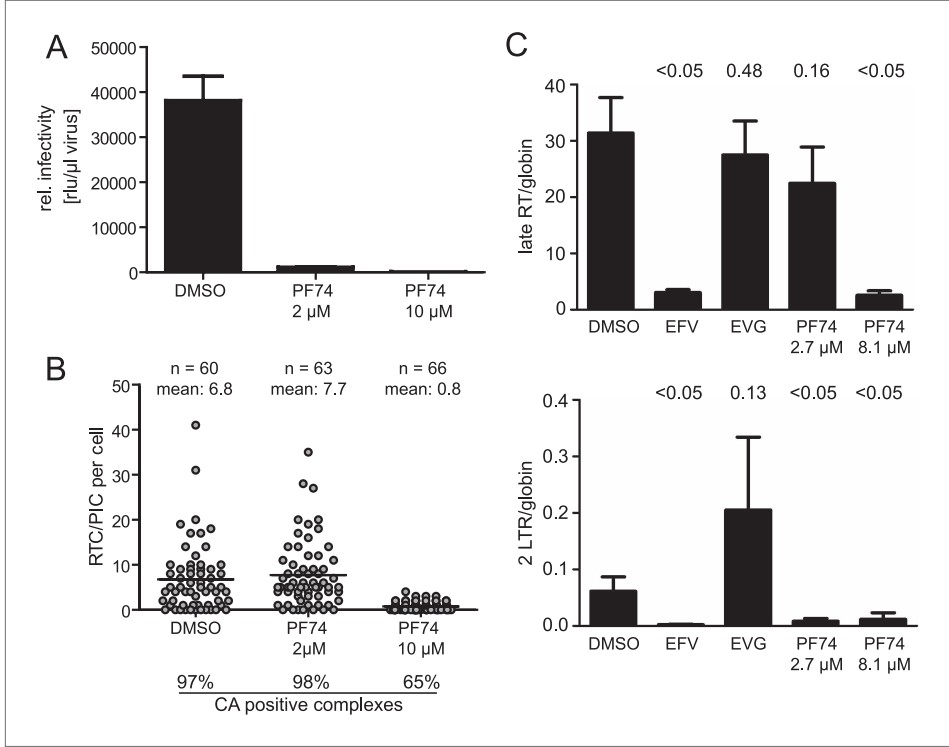

**Figure 5**. Effect of PF74 treatment on RTC/PIC formation and infection of host cells. (**A**) Inhibition of productive infection. TZM-bl cells were infected with serial dilutions of HIV-1 produced in the presence of DMSO or the indicated concentrations of PF74, respectively. Cell lysates were harvested at 48 hr p.i., and infectivity was determined by measuring luciferase reporter gene activity. The graph shows mean values and standard deviation (SD) from three replicate infections. (**B**) Inhibition of RTC/PIC formation. TZM-bl cells were infected with HIV-1 (IN.eGFP) in the presence of DMSO or two different concentrations of PF74, respectively. At 4.5 hr p.i., cells were fixed, click labeled, immunostained with antiserum against HIV-1 CA, and analyzed. Numbers of RTC/PIC per cell were determined for randomly selected cells from z-stacks covering the whole-cell volume, and the proportion of complexes co-localizing with CA immunostaining was determined. The figure shows pooled results from three independent experiments. Statistical significance was assessed using the Mann–Whitney test. (**C**) Quantification of HIV-1 DNA. MT-2 cells were infected with HIV-1 (IIIB) in the presence of DMSO or of the indicated concentrations of the inhibitors EFV, EVG, or PF74, respectively. Late RT products (top) and 2-LTR circles (bottom) in cell lysates were quantified by qPCR as described in 'Materials and methods' using appropriate primer sets. The graph shows mean values and SD from three replicate infections. Statistical significance of the difference between each treatment and the DMSO control (p-values determined by Student's t test) is indicated.

macrophages is more specific, but much less efficient than in HeLa-derived cells. Furthermore, the cellular restriction factor SAMHD1 inhibits HIV-1 replication in primary macrophages at the stage of reverse transcription (**Hrecka et al., 2011**). Accordingly, the average number of RTC/PIC detected in macrophages observed at 24 hr or 48 hr after infection was 10- to 50-fold lower compared to HeLa-derived cells infected for 4 hr and showed significant donor dependent variation (**Figure 7—source data 1**). Analyzing a total of 567 MDM from eight different donors in five independent experiments, we observed a total of 135 RTC/PIC in the cytoplasm of these cells. HIV-1 CA was found to be associated with only 44% of these RTC/PIC (**Table 1**), while >97% of cytoplasmic RTC/PIC were CA-positive in the case of TZM-bl cells. The proportion of CA-positive RTC/PIC in MDM decreased over time with ~50% CA-positive at 24 hr and ~30% CA-positive at 48 hr (**Table 1**), indicating time-dependent loss of CA from cytoplasmic RTC/PIC.

As post-mitotic cells, MDM lack nuclear DNA synthesis and therefore exhibited almost no nuclear EdU signal. Lack of cellular DNA synthesis in the nucleus greatly facilitated visualization of nuclear HIV-1 PIC in these cells (**Figure 7B,C**), and we detected 70 nuclear PIC in the total of 567 MDM that were analyzed in this study. The percentage of nuclear PIC (over all RTC/PIC detected) increased

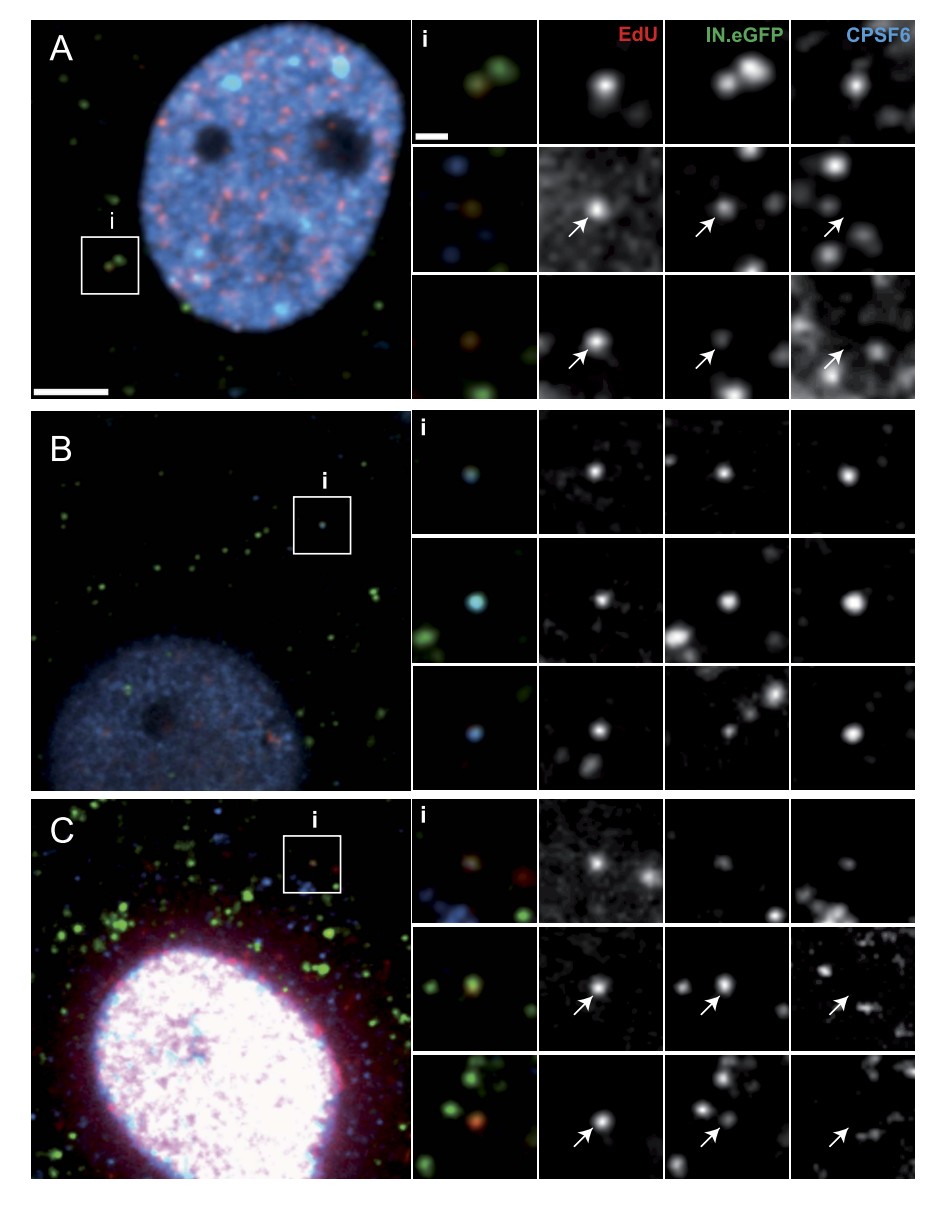

**Figure 6**. Co-localization of CPSF6 with RTC/PIC in the cytoplasm. The figure shows co-localization of CPSF6 with RTC/PIC in the cytoplasm of TZM-bl (**A**) and TNPO3KD (**B** and **C**) cells in the absence (**B**) or presence (**C**) of 2 μM PF74. Cells were infected with HIV-1 (IN.eGFP), and click-labeled as described in *Figure 1A*, followed by immu-nostaining against CPSF6 (blue) and SDCM analyses. The RTC/PIC in the boxed area (**i**) are shown enlarged in the top rows at the right; the middle and bottom rows at the right show examples of RTC/PIC from other cells. Arrows in enlargements indicate positions of RTC/PIC without detectable CPSF6. To visualize weak signals for proper visualization, greyscale enlargement images were auto-contrasted in ImageJ. Scale bars: 5 μm (overviews), 1 μm (enlargements). See *Figure 6—figure supplement 1* for localization of CPSF6 in different cell lines.

The following figure supplement is available for figure 6:

**Figure supplement 1**. CPSF6 localization in TZM-bl (**A**), TNPO3Scr (**B**), and TNPO3KD (**C**) cells.

from 9% at 24 hr to 58% at 48 hr (*Table 1*). Surprisingly, almost all nuclear PIC (68 of 70) exhib-ited a clearly detectable CA signal (*Figure 7B,C*; *Table 1*), suggesting that the intact capsid or a capsid-derived structure remains associated with the replication complex during and after nuclear import.

**Table 1.** Cytoplasmic and nuclear subviral complexes in MDM

| | Cytoplasm | | | Nucleus | | | Proportion of nuclear complexes [%] | Number of cells imaged |
|---|---|---|---|---|---|---|---|---|
| | RTC/PIC (n) | CA-positive RTC/PIC (n) | CA-positive RTC/PIC [%] | PIC (n) | CA-positive PIC (n) | CA-positive PIC [%] | | |
| **24 hr** | 90 | 46 | **51** | 9 | 9 | **100** | 9 | 367 |
| **48 hr** | 45 | 13 | **29** | 61 | 59 | **97** | 58 | 200 |
| **Total** | 135 | 59 | **44** | 70 | 68 | **97** | 34 | 567 |

The table summarizes data from primary MDM infected with NL4-3-R5(IN.eGFP) or NL4-3-4059(IN.eGFP) for 24 hr or 48 hr, respectively. Infected MDM were immunostained with antibodies against nuclear pore complexes and CA. Data were obtained as outlined in *Figure 7* in cells from eight different donors in five independent experiments (cells from three donors were infected for both 24 hr and 48 hr).

## Discussion

In this study, we describe a robust strategy for detecting and characterizing productive HIV-1 post-entry complexes by identifying objects with nascent DNA synthesis co-localizing with viral replication proteins in the cytoplasm or in the nucleus of infected cells. EdU click-labeling has no specificity for viral DNA, but the observation of IN.eGFP/EdU-positive objects in the extranuclear region, not over-lapping with mitochondria and almost completely blocked by EFV, clearly identified functional HIV-1 RTC/PIC in proliferating reporter cells. Nuclear PIC were only detectable in cells that had not under-gone DNA synthesis during the labeling period in several HeLa-derived cell lines, while lack of cellular DNA synthesis in post-mitotic macrophages allowed easier detection of nuclear PIC in this case. EdU-positive HIV-1 complexes mainly accumulated between 2 hr and 4 hr after infection of HeLa-derived cells, consistent with detection of reverse transcription products by quantitative PCR in this cell type at early time points (*Thomas et al., 2011*). RTC/PIC were detected much later upon infection of pri-mary MDM and accumulated over time; this slower formation is again consistent with previous PCR results (*Ambrose et al., 2012*). Furthermore, nuclear PIC mainly accumulated between 24 hr and 48 hr in MDM, indicating the time frame of nuclear import in this cell type. In contrast to PCR-based detection of RT products or nuclear 2-LTR circles, which has to be performed on bulk extracts, the imaging method established here allows identification of individual RTC/PIC in the physiological envi-ronment of the infected cell.

EdU is incorporated in the position of thymidine in the viral cDNA, and we can thus expect a maximum of 5588 EdU molecules for the entire double-stranded cDNA of HIV-1 (NL4-3). Obviously, the actual number is expected to be much lower as EdU competes with the pool of cellular dTTP. Our experimental system operates at almost single molecule sensitivity (detection limit 1–2 dye mol-ecules per confocal volume; *Figure 1—figure supplement 6A*), but RTC/PIC detection also depends on relative EdU incorporation and click-labeling efficiency. Using lentiviral vectors of different lengths, we could identify nascent cDNA for vector genomes of 5.4 kb and 3.7 kb (*Figure 1—figure supple-ment 6B*), indicating that detection of short reverse transcription products, as suggested for abor-tive infection of resting primary T-cells (*Doitsh et al., 2014*), may be possible with this method. Accumulation of RTC/PIC over several hours may reflect asynchronous cell entry and initiation of reverse transcription in combination with the relatively slow process of minus-strand cDNA synthesis in infected cells (*Thomas et al., 2007a*).

The vast majority of cytoplasmic RTC/PIC in HeLa-derived cells were associated with the viral structural proteins CA and NC but lacked MA. As a membrane-associated protein, MA can be expected to separate from the incoming capsid upon virus–cell fusion, but some previous studies reported residual MA retained on HIV-1 RTC (*Bukrinsky et al., 1993*; *Gallay et al., 1995*; *Miller et al., 1997*; *Iordanskiy et al., 2006*). Clearly, lack of MA detection by immunofluorescence does not rule out the presence of some MA molecules on RTC/PIC, but the bulk of MA appears to be lost. NC is a nucleic-acid binding protein that coats the viral RNA genome, and NC mutations have been shown to affect HIV-1 integration (*Cimarelli et al., 2000*; *Thomas et al., 2011*). The association of NC with almost all RTC/PIC may thus not be surprising, but NC had not been detected in several studies where RTC and/or PIC were biochemically fractionated (*Farnet and Haseltine, 1991*; *Miller et al., 1997*).

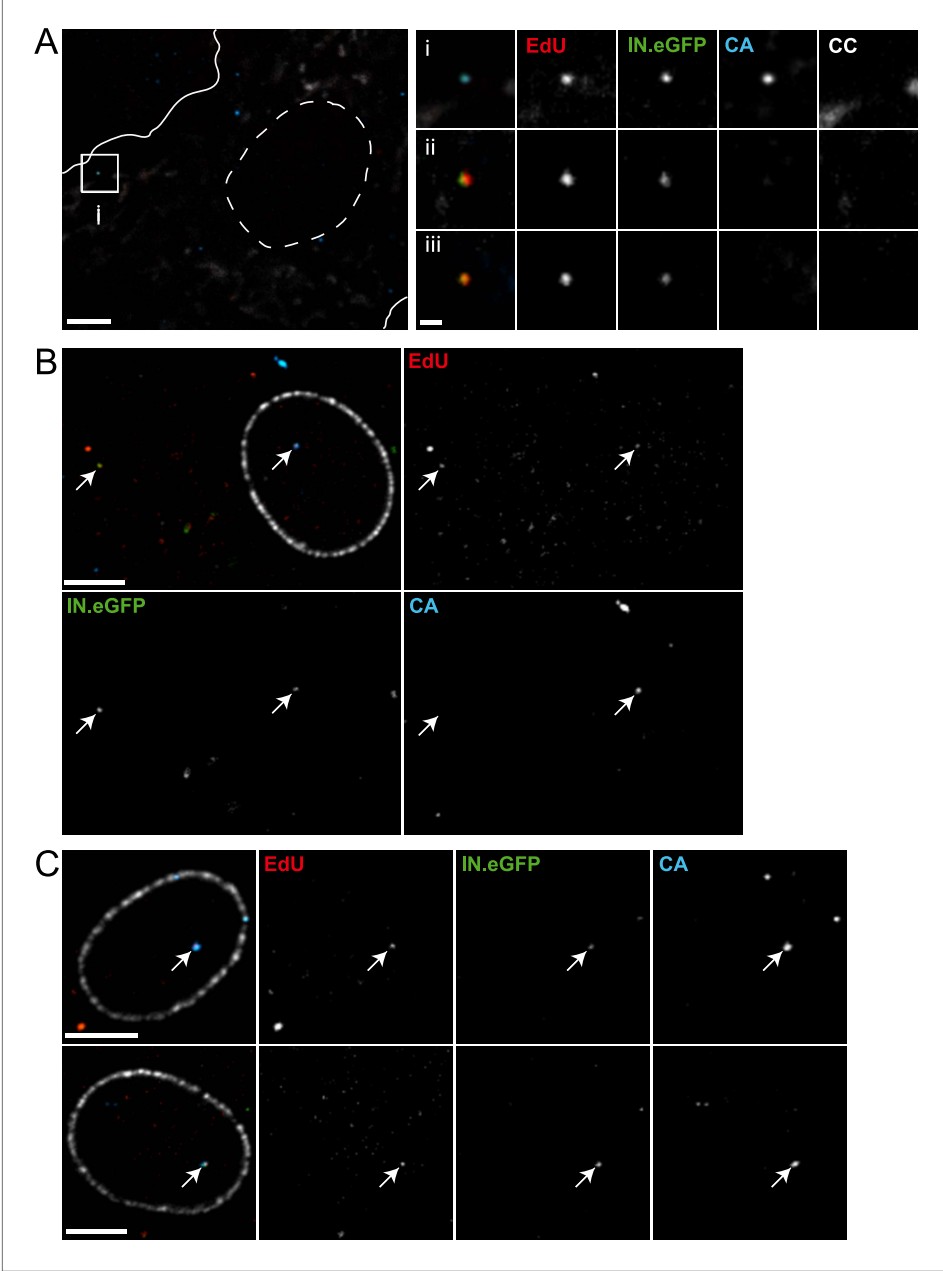

**Figure 7**. Detection of RTC/PIC in primary macrophages. (**A**) Detection of RTC/PIC in MDM. Human MDM were prepared and infected with an R5-tropic variant of HIV-1 (IN.eGFP) (NL4-3-R5(IN.eGFP)) for 24 hr in the presence of 5 μM EdU. Cells were fixed, click-labeled, and immunostained for CC and HIV-1 CA. The left panel shows a z-section through part of a representative cell, displaying immunostaining for CC (white) and HIV-1 CA (blue) together with EdU (red) and IN.eGFP (green). Boundaries of the cell (white line) and nucleus (dashed line) are indicated. The RTC/PIC in the boxed area (**i**), as well as two other exemplary RTC/PIC detected in cells from two other donors (**ii**–**iii**) are shown enlarged on the right. To visualize weak signals for proper visualization, greyscale enlargement images of example (**i**) were auto-contrasted in ImageJ. Scale bar: 5 μm (overview), 1 μm (enlargements). (**B** and **C**) CA signals associated with nuclear PIC. MDM were prepared and infected for 48 hr, followed by click-labeling, and immunostaining using an antibody against nuclear pore complexes (white) and antiserum against HIV-1 CA (blue). (**B**) Channel overlay of a z-section through part of a representative cell, together with images of the individual channels (EdU, red; IN.eGFP, green; CA, blue). Arrows indicate an example of a cytoplasmic RTC/PIC (left) and a nuclear PIC (right), respectively. Scale bar: 5 μm. (**C**) Additional examples of nuclear PIC

*Figure 7. Continued on next page*

*Figure 7. Continued*

detected at 48 hr p.i. in MDM from two other donors. Scale bars: 5 μm. See *Figure 7—source data 1* for a summary of infected MDM immunostained with CC and CA antibodies.

The following source data is available for figure 7:

**Source data 1**. RTC/PIC detected in MDM.

Association of almost all cytoplasmic RTC/PIC with CA in HeLa-derived cells clearly indicates that reverse transcription occurs in the viral capsid or a capsid-derived structure in these cells. CA association of cytoplasmic RTC/PIC was independent of subcellular localization, arguing against a gradual loss of CA during trafficking to the nuclear pore. Consistent with this hypothesis, we observed a similar or slightly higher CA labeling intensity on RTC/PIC compared with EdU-negative intracellular HIV-1 particles. Several previous studies reported a time-dependent loss of CA signals on cytoplasmic structures positive for the HIV-1 accessory protein Vpr in HeLa- or HOS-derived cell lines infected with VSV-G pseudotyped VLPs (*McDonald et al., 2002*; *Hulme et al., 2011*), suggesting gradual capsid uncoating during reverse transcription. Alternatively, the reported loss of CA signal may have been due to the inaccessibility of the reactive epitope due to conformational changes and/or host protein binding after cytoplasmic entry, especially since monoclonal antibodies were used in these studies. An influence of the entry pathway may also be considered as these prior studies used VSV-G pseudotyped particles entering via the endosomal route. RTC/PIC co-localizing with nuclear pore proteins in TZM-bl cells also exhibited a strong CA signal, while CA was absent from all but one nuclear PIC indicating that CA is lost at the nuclear pore or shortly after nuclear entry in this cell type. This hypothesis is consistent with the described interaction of CA with nucleoporins 153 and 358 (*Schaller et al., 2011*; *Matreyek et al., 2013*) and suggests that cDNA within the intact viral capsid or a capsid-derived structure docks to the nuclear pore complex followed by dissociation of CA and nuclear import.

A very different picture was observed for CA association of RTC/PIC in primary human MDM. Less than half of the RTC/PIC showed a detectable CA signal in this case, while >97% of all RTC/PIC were CA-positive in HeLa-derived cells. This difference is unlikely to be caused by earlier CA uncoating of productive RTC/PIC in MDM, since almost all nuclear PIC (97%) were CA-positive in this cell type. We even observed a CA-positive nuclear PIC and a CA-negative cytoplasmic RTC/PIC in the same cell (*Figure 7B*). These observations suggest that the intact or remodeled viral capsid is imported through the intact nuclear pore in post-mitotic MDM following complete or partial reverse transcription. This is completely different from the results observed in HeLa-derived cells and is not predicted by any of the current models for early HIV-1 replication. The fact that nuclear PIC in MDM were clearly CA-positive, while the majority of cytoplasmic RTC/PIC were CA-negative in the same cells, and CA appeared to be lost over time suggests that many of the cytoplasmic RTC/PIC detected in MDM were not predecessors of nuclear PIC but may rather represent dead-end products. These (at least partially) uncoated cytoplasmic structures contain incomplete or complete HIV-1 cDNA, which can be exposed to cytoplasmic DNA sensors. They may thus contribute to the strong innate immune response in MDM (*Lahaye et al., 2013*; *Rasaiyaah et al., 2013*). It is difficult to envision, however, how the CA-negative cytoplasmic RTC/PIC could give rise to nuclear PIC, since almost all nuclear PIC were strongly CA-positive. Our results therefore suggest that the capsid or a capsid-derived structure remains intact not only during reverse transcription but also during nuclear import, at least in MDM. Lack of a CA signal on nuclear PIC in the proliferating HeLa-derived reporter cell line may indicate the loss of the capsid at the nuclear pore, but prior to nuclear import, or may be due to more rapid removal of CA after nuclear import in these cells compared to MDM. It appears likely that the capsid has to be removed before proviral integration in all productively infected cells, and the two CA-negative nuclear PIC in MDM in the present study may thus reflect a later stage after nuclear entry. Accordingly, CA-negative nuclear PIC were only detected 48 hr after infection, while all nuclear PIC were CA-positive at 24 hr after infection. Detailed investigation of the time course of events will require detection of RTC/PIC in living cells, however, which is not possible with the current experimental system.

Imaging productive HIV-1 RTC/PIC allowed us to unravel the dose-dependent bimodal inhibitory mechanism of the CA-directed antiviral PF74 and to directly prove association of CPSF6 with RTC/PIC. Consistent with previous reports (*Shi et al., 2011*), we observed loss of reverse transcription products

and of RTC/PIC when infecting cells in the presence of 10 µM PF74. This concentration is much higher than the $IC_{90}$ of PF74, however, and inhibition at lower concentrations, while still blocking HIV-1 infection, did not affect RTC/PIC formation or CA association, but rather blocked nuclear import. PF74 binds the same pocket in the N-terminal domain of CA as the cellular proteins CPSF6 and Nup153, both of which have been implicated in nuclear import of the PIC (*Matreyek et al., 2013*). Recently, Price et al. revealed that the CA hexamer of the mature lattice (and not monomeric CA) constitutes the optimal interface for Nup153 and CPSF6, which have overlapping binding sites on the capsid that are lost upon disassembly (*Price et al., 2014*). It appears likely, therefore, that PF74 acts as a competitive inhibitor of host factor binding on the intact (or semi-intact) capsid structure at limiting drug concentrations, thereby blocking CPSF6 association and nuclear import. Accordingly, we observed a strong reduction of CPSF6 association with RTC/PIC in the presence of 2 µM PF74 when cytoplasmic CPSF6 concentration was increased by knock-down of its nuclear import factor TNPO3. Saturating concentrations of PF74 may lead to dissociation of the entire capsid structure with consequent loss of reverse transcription, but this does not appear to be the primary mode of action of this antiviral.

Many host cell proteins including Trim5α, cyclophilin A, CPSF6, MX-2, TNPO3, as well as nucleoporins have been implicated in early HIV-1 post-entry stages in a CA-dependent way (*Matreyek and Engelman, 2013*; *Ambrose and Aiken, 2014*; *Hilditch and Towers, 2014*). CA mutations affecting many of these interactions are available, and differential effects of these mutations on infectivity have been reported depending on the host cell (*Matreyek and Engelman, 2013*). Direct identification of individual productive HIV-1 RTC/PIC by microscopy will be ideally suited to determine the role of these factors in HIV-1 post-entry stages in various host cells including MDM. The association of nuclear PIC with CA in primary human MDM challenges the current hypothesis concerning early HIV-1 replication and shows the power of this direct imaging-based approach. Detection of productive RTC/PIC and their association with viral and host factors can thus be expected to find wide applications in the future and will contribute to unraveling the current mysteries of HIV-1 post-entry events.

## Materials and Methods

### Cells and plasmids

TZM-bl cells (*Wei et al., 2002*) and human embryonic kidney 293T cells were grown at 37°C in the presence of 5% $CO_2$ in Dulbecco's modified Eagle's medium (DMEM; Life Technologies, Germany), supplemented with 10% fetal calf serum (FCS; Biochrom, Germany), 100 U/ml penicillin, and 100 µg/ml streptomycin. HeLaP4 derived cell lines stably transduced with shRNA targeting TNPO3 (referred to here as TNPO3KD) or a scrambled shRNA control (referred to here as TNPO3Scr), respectively, were kindly provided by Zeger Debyser (University of Leuven, Belgium) (*Thys et al., 2011*). Cells were propagated in complete DMEM supplemented with 1 µg/ml puromycin (Life technologies). MT-2 cells (*Harada et al., 1985*) were grown in RPMI medium with the same supplements. Human peripheral blood mononuclear cells (PBMCs) were isolated from buffy coats of healthy blood donors by Ficoll density gradient centrifugation. PBMCs were seeded in 8-well LabTek chamber slides (Thermo Fisher Scientific, Waltham, MA) in DMEM containing 10% heat inactivated FCS and incubated at 37°C for 6 hr. Subsequently, floating lymphocytes were removed and adherent monocytes were washed followed by cultivation in DMEM containing 10% heat inactivated FCS and 10% human AB serum (Sigma Aldrich, St. Louis, MO) for 7 days to allow differentiation into macrophages.

The HIV-1 proviral plasmid pNL4-3 has been described (*Adachi et al., 1986*). Plasmid pNL4-3-delEnv carries a 2 bp fill-in in the *env* ORF resulting in a frameshift and premature stop codon. Plasmid pNLC4-3-R5 was cloned by exchanging a StuI/XhoI fragment comprising part of the Env coding region with the corresponding fragment from plasmid pCAGGS.NL4-3R5 which carries mutations in the V3-loop coding region conferring CCR5 tropism (*Bozek et al., 2012*). Plasmid pEnv-4059 expressing an Env protein from a primary HIV-1 isolate (*Schnell et al., 2011*) was kindly provided by R Swanstrom (University of North Carolina, USA). Plasmid pVpr.IN.eGFP (*Albanese et al., 2008*) encoding a Vpr.IN.eGFP fusion protein with an HIV-1 protease recognition site between Vpr and IN was kindly provided by Anna Cereseto (CIBIO, Mattareo, Italy). To generate plasmid pVpr.IN.mEos3.2, a BamHI/NotI fragment of pVpr.IN.eGFP comprising the eGFP coding region was replaced with a PCR fragment of the mEos3.2 coding sequence from pCMVmEos3.2 (*Zhang et al., 2012*) flanked by BamHI/NotI cleavage sites. PCR primers used were: forward primer CGCGGATCCACCGGTCGCCACCATGAGTGCGATTAAGCCAG; reverse primer CGCGCGGCCGCTTATCGTCTGGCATTGTCAG. Plasmid

pRRL.PPT.SF.GFPpre (*Schambach et al., 2006*) was kindly provided by Jens Bohne (Hannover Medical School, Germany). Plasmid pAdVAntage was from Promega, plasmids pWPI, pMD2.G, and psPAX2 were generated in the lab of Didier Trono (EPFL, Lausanne, Switzerland) and obtained through Addgene.

## Antisera and reagents

Rabbit polyclonal antisera against HIV-1 CA, MA, NC, or IN were raised against purified recombinant proteins. Mouse monoclonal antibody 414, recognizing phenylalanine–glycine repeats of nuclear pore complex proteins, was from Abcam (UK); mouse monoclonal antibody against cytochrome C was from BD Biosciences (Franklin, NJ); affinity purified rabbit antibody against CPSF6 was obtained from Sigma (HPA039973). Secondary antibodies Alexa Fluor 405 Goat Anti-Mouse IgG, Alexa Fluor 532 Goat Anti-Rabbit IgG, and Alexa Fluor 568 Goat Anti-Rabbit IgG were from Life Technologies.

Stock solutions of 6 mM aphidicolin (Sigma A0781), 5 mM efavirenz (obtained through the AIDS Research and Reference Reagent Program, Division of AIDS, NIAID), 10 mM PF74 (Gilead Sciences, Foster City, CA), and 10 mM elvitegravir (Gilead Sciences), respectively, were prepared in dimethyl sulfoxide (DMSO) and stored at −20°C.

## Virus production and characterization

HEK 293T cells were transfected with pNL4-3 or pNLC4-3-R5 (for producing NL4-3-R5(IN.eGFP)), respectively, using a standard $CaPO_4$ transfection procedure. For producing viruses containing IN.eGFP or IN.mEos3.2, the respective expression vector was co-transfected with the proviral plasmid at a molar ratio of 1:4.5. R5-tropic virus (NL4-3-4059(IN.eGFP)) carrying Env-4059 from a primary patient isolate (*Schnell et al., 2011*) was produced by co-transfecting HEK293T cells with pVpr.IN.eGFP, pEnv-4059, and pNL4-3-delEnv in a molar ratio of 1:1:4.5. For production of lentiviral vector particles, cells were transfected with pMD2.G, psPAX2, pAdVAntage, and the lentiviral vector pRRL.PPT.SF.GFPpre or pWPI, respectively, using a molar ratio of 2.8:2.8:1:4. Virus or vector particle containing supernatants were harvested at 36 hr post-transfection, filtered through 0.45 µM filters and virus was concentrated by ultracentrifugation through a 20% (wt/wt) sucrose cushion. Particles were resuspended in PBS containing 10% FCS and 10 mM HEPES (pH7.5) and stored in aliquots at −80°C. For immunoblot analyses, samples were separated by SDS-PAGE (12.5%) and proteins were transferred to a polyvinylidene-difluoride membrane by semi-dry blotting. Detection was performed using a LiCor Odyssey instrument, using the indicated primary antisera with corresponding LiCor secondary antibodies.

Relative infectivity of virus was analyzed by titration on TZM-bl indicator cells. At 48 hr post-infection, cells were lysed and luciferase activity was quantitated using the SteadyGlo assay kit (Promega) according to the manufacturer's instructions. Values obtained were normalized to the amounts of virus particles as assessed by p24 ELISA using an in-house protocol. Determination of virus titers was performed by titration on TZM-bl indicator cells followed by detection of beta-lactamase expressing cells as described previously (*Wei et al., 2002*).

## PCR quantification of reverse transcription products

MT-2 cells were infected at an m.o.i. of 10 with HIV-1 IIIB pelleted from infected H9 cells (Advanced Biotechnologies, Eldersburg, MD; catalog number 10-124-000) in the presence of 6 µg/ml polybrene by mutation for 3 hr at 37°C. Cells were washed and seeded onto 6-well plates (1 × 10⁶ per well) in a total volume of 2 ml RPMI medium supplemented with FCS. DMSO or inhibitors were added and cells were incubated at 37°C for 12 hr for late RT product quantification or 24 hr for 2-LTR circle product quantification, respectively. Viral DNA was isolated using a QIAamp DNA mini kit (Qiagen, Germany) and quantified using TaqMan real-time PCR using the ABI Prism 7900HT sequence detection system (Life Technologies). Primer-probe sets used were: Late RT products, PBS-F (5'- TTTTAGTCAGTG TGGAAAATCTCTAGC-3'), PBS-R (5'-TTGGCGTACTCACCAGTCGCC-3'), and PBS probe (5'-6FAMTCGACGCAGGACTCGGCTTGCT-6TAMSp-3'). Primer-probe sets for 2-LTR circles and the host β-globin gene (used to normalize for cell number) were as previously described (*Butler et al., 2001*).

## Infectivity analysis by flow cytometry

2 × 10⁵ and 4 × 10⁵ TZM-bl cells per well were seeded in 6-well plates and pre-incubated at 37°C with complete DMEM containing DMSO or 6 µM APC for 24 hr. Cells were infected with HIV-1 or HIV-1 (IN.eGFP), respectively, with an m.o.i. of 0.1 in the presence of DMSO or 6 µM APC. At 4 hr p.i., medium was replaced by DMEM and incubation was continued. At 48 hr p.i., cells were fixed with 3%

PFA, permeabilized and probed with phycoerythrin-conjugated monoclonal anti-HIV CA antibody KC57-RD1 (Beckman Coulter, Germany). The proportion of infected cells was quantified by flow cytometry using a BD FACSVerse instrument.

## Virus infection and EdU click-labeling

TZM-bl cells were seeded in 8-well LabTek chamber slides in DMEM, 10% FCS containing 6 µM APC. On the following day, virus infection was performed at an m.o.i. of 25 in the same medium composition and 10 µM EdU (Life Technologies) was added. Cells were pre-incubated at 16°C for 30 min and then shifted to 37°C for 2 hr. Subsequently, infection was either stopped, or the infection mixture was replaced by pre-warmed medium containing 6 µM APC and 10 µM EdU, and incubation at 37°C was continued for the specified time periods. To stop infection, cells were washed with PBS and fixed with 3% paraformaldehyde (PFA) at room temperature for 30 min. Cells were washed and permeabilized with 0.2% (vol/vol) Triton X-100. Click-labeling was performed for 40 min at room temperature using the Click-iT EdU-Alexa Fluor 647 Imaging Kit (Life Technologies) according to the manufacturer's instructions, followed by immunostaining with the indicated antisera.

In the case of MDM infection, APC incubation was omitted and infection was performed in the presence of 5 µM EdU. Virus corresponding to 75 ng CA was added to each well and incubated with cells for 24 hr or 48 hr. The amount of virus particles applied yielded an m.o.i. of 50 or 25 on TZM-bl cells for NL4-3-R5(IN.eGFP) or NL4-3-4059(IN.eGFP) viruses, respectively. Cell-fixation, click-labeling, and immunofluorescence staining were performed as described for TZM-bl cells.

## Immunostaining and quantitation of signal intensity

Fixed and permeabilized cells were blocked with 3% BSA–PBS, washed and incubated with the respective first and secondary antisera for 1 hr each at room temperature. Signal quantification was performed using ImageJ. Equal amounts of objects representing cytoplasmic RTC/PIC or viral particles lacking EdU signals were randomly selected from images taken in three independent experiments. Background signal measured in the extracellular area was subtracted in each channel. Sum signal intensity was calculated for each object in the CA and IN.eGFP channels. Signal intensities of CA or IN.eGFP of RTC/PIC were normalized for the mean values of CA or IN.eGFP obtained from all CA–IN.eGFP objects lacking EdU labeling from the same experiment.

## Spinning disc confocal microscopy

Multi-channel 3D image series were acquired with a PerkinElmer UltraVIEW VoX 3D SDCM using a 60× or 100× oil immersion objective (NA 1.49) (Perkin Elmer, Waltham, MA), with a z-spacing of 200 nm. Images were recorded in the 405, 488, 561, and 640 nm channels. For quantitation of co-localizing objects, cells were randomly selected for imaging. Co-localization of signals detected in different channels was determined manually on each plane of the image series. Data were analyzed using GraphPad Prism.

## Super-resolution microscopy

Super-resolution microscopy was performed using a custom-built microscope setup described earlier (*Nanguneri et al., 2012*). Briefly, a multi-line argon–krypton laser (Innova70C, Coherent, Santa Clara, CA) and a 405 nm diode laser (Cube, Coherent) were coupled into an inverted microscope (IX71, Olympus) equipped with a 63× oil immersion objective (PlanApo 63×, NA 1.45, Olympus) suitable for total internal reflection fluorescence (TIRF) imaging. The excitation and emission beams were separated using appropriate dichroic mirrors and filters (AHF, Germany). The fluorescence emission was detected by an EM-CCD camera (Ixon, Andor, UK).

PALM/dSTORM imaging was performed using an imaging buffer which is suitable for both photoswitching the fluorescent protein mEos3.2 as well as the organic fluorophores Alexa Fluor 532 or Alexa Fluor 647 (*Endesfelder et al., 2011*). Briefly, the cells were imaged in oxygen-depleted hydrocarbonate buffer (pH 8) supplemented with 50 mM mercaptoethylamine (MEA). First, Alexa Fluor 647 was reversibly photoswitched by irradiation with 488 nm (photoactivation) and 647 nm (read-out). For each channel, 8000 images were recorded with an integration time of 50 ms. Then mEos3.2 was photoactivated by irradiation with 405 nm and imaged using an excitation wavelength of 568 nm. Alexa Fluor 532 was photoactivated by irradiation with 514 nm. Single-molecule localization and image reconstruction was performed using the rapidSTORM software (*Wolter et al., 2011*). The localization accuracy of single-molecule super-resolution microscopy was evaluated experimentally as described earlier using a custom written software (coordinate based localization precision estimator,

provided as supplement to *Endesfelder et al. (2014)*). For each localized fluorophore, the distance to its nearest neighbor fluorophore in an adjacent frame was calculated. As the majority of fluorophores are detected in multiple adjacent frames, the maximum of the nearest neighbor distance distribution represents the error of localization. A pre-requisite for using this approach is a statistically significant number of events (n ~ 4000). Images were recorded by adding multi-spectral beads (Life Technologies) to the sample and post-aligning the individual images (*Malkusch et al., 2012*).

## Acknowledgements

We thank Anna Cereseto (CIBIO, Mattareo, Italy) for kindly providing plasmid pVpr.IN.eGFP, Pingyong Xu (Chinese Academy of Sciences, Bejing, China) for plasmid pCMVmEos3.2, Ronald Swanstrom (University of North Carolina, USA) for plasmid pEnv-4059, Jens Bohne (Hannover Medical School, Germany) for plasmid pRRL.PPT.SF.GFPpre, and Manon Eckhardt for plasmid pNLC4-3R5. EFV was obtained through the AIDS Research and Reference Reagent Program, Division of AIDS, NIAID. TNPO3KD and TNPO3SCR control cells were kindly provided by Zeger Debyser (University of Leuven, Belgium). We gratefully acknowledge Mike Heilemann (Goethe University, Frankfurt, Germany) for helpful suggestions and for support in PALM/dSTORM experiments. We thank Maria Anders for expert technical assistance. We gratefully acknowledge Meinhard Kieser (Institute for Medical Biometry, Heidelberg University Hospital) for help with statistical analysis.

This work was supported in part by a grant from the Deutsche Forschungsgemeinschaft in SFB1129 and by the German Center for Infection Research (DZIF; project 7.5 TTU HIV). KP was supported by a postdoctoral fellowship from the Medical Faculty Heidelberg. HGK and BM are investigators of the CellNetworks Cluster of Excellence (EXC81).

## Additional information

### Funding

| Funder | Grant reference number | Author |
|---|---|---|
| Deutsche Forschungsgemeinschaft | SFB1129 | Hans-Georg Kräusslich |
| Helmholtz Association | German Centre for Infection Research - Project 7.5 TTU | Hans-Georg Kräusslich |
| University of Heidelberg | Postdoc stipend | Ke Peng |

The funders had no role in study design, data collection and interpretation, or the decision to submit the work for publication.

### Author contributions

KP, Conception and design, Acquisition of data, Analysis and interpretation of data, Drafting or revising the article; WM, BG, VL, Acquisition of data, Analysis and interpretation of data; SRY, Acquisition of data, Contributed unpublished essential data or reagents; LT, TC, Analysis and interpretation of data, Contributed unpublished essential data or reagents; BM, Analysis and interpretation of data, Drafting or revising the article; H-GK, Conception and design, Analysis and interpretation of data, Drafting or revising the article

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
