## [Decision Letter]

Thank you for sending your work entitled “Quantitative microscopy of functional HIV post-entry complexes reveals association of replication with the viral capsid” for consideration at *eLife*. Your article has been favorably evaluated by Vivek Malhotra (Senior editor) and 3 reviewers, one of whom is Wesley Sundquist, a member of our Board of Reviewing Editors.

The Reviewing editor and the other reviewers discussed their comments before we reached this decision, and the Reviewing editor has assembled the following comments to help you prepare a revised submission.

The authors describe a method for imaging single HIV-1 core particles as they traverse the cytoplasm and enter the nucleus. The approach employs a clever use of nucleotide/click chemistry to distinguish cores that are actively reverse transcribing their genomes from the bulk of cores that are inactive and/or have been taken up by endocytosis. As the authors point out, there is currently a great need for experimental approaches for characterizing the early post-entry stages of HIV replication, and this study is important in that regard. In addition, the work is well presented and the characterization of the methodology is generally thorough and convincing, giving confidence that authentic HIV cores really are being visualized. These are all significant strengths. We are concerned, however, that the current study does not yet go far enough in applying the methodology to discover new biology, that the experiments performed in macrophages are not yet conclusive, and that the imaging data need to be more quantitative and the (lack of) effects of APC treatment need to be documented more thoroughly.

The authors attempt to answer two different questions: 1) when does the RTC/PIC complex uncoat (i.e., lose the bulk of the CA subunits that constitute the capsid), and 2) what is the mechanism of action of the PF74 inhibitor. The first question is a very important one mechanistically, and the authors do a nice job of demonstrating that in HeLa cells, the bulk of the CA subunits remain associated until the RTC/PIC complex approaches or engages the nuclear pore. The data are the most definitive yet in support of the increasingly attractive idea that reverse transcription and cytoplasmic transport occur within an intact (or nearly intact) capsid. A shortcoming, however, is that the authors perform analogous studies in macrophages (which is to their credit), but then fail to draw any definitive conclusions in this relevant cell type, instead arguing that “the number of detectable objects was relatively low in most cases, however, and quantitative analysis of a larger number of RTC/PIC in MDM from various donors will be required to determine whether CA is lost more easily from RTC/PIC in these cells.” The authors deserve credit for not overinterpreting limited data, but need to perform sufficient numbers of MDM experiments to allow definitive conclusions to be drawn.

Experiments aimed at defining the mechanism of action of PF74 indicate that at IC50-relevant concentrations, the drug acts by binding intact capsids and inhibiting interactions of relevant host cofactors. The data are strong as far as they go, but the inhibition of host factor interactions and the putative block to nuclear import is really inferred by process of elimination (i.e., reverse transcribing, intact capsids are observed normal levels in the presence of 2 microM PF74, as assayed in imaging and qPCR experiments, yet 2-LTR circle formation is inhibited, as assayed by qPCR, so the authors conclude that there must be a block to nuclear import). In addition, the capsid uncoating assay wasn't really required to conclude that RT is normal but 2-LTR circle formation is blocked (though, of course, it was useful to see that the capsids remained intact under these conditions). Moreover, one could argue that defining the mechanism of PF74 action isn't really a cutting edge problem, particularly given a recent series of nice papers from the Engelman, James and other labs that strongly imply that PF74 binding must inhibit interactions with host factors like CPSF6 and nucleoporins.

Major issues:

1) The authors should investigate HIV-1 capsid uncoating in macrophages to the point where they can draw definitive conclusion(s). This will require imaging sufficient numbers of core particles for capsid content, and may also involve the use of other tricks to examine the status of RTC/PIC complexes (e.g., SAMHD1 depletion or Vpx treatment). These are admittedly challenging experiments, but without them the current study does not resolve the current controversy over where and when capsid uncoating occurs.

2) The authors should demonstrate that their methodology can be used to examine RTC/PIC interactions with host factors in a meaningful way. This could be done by performing imaging experiments that demonstrate directly that PF74 inhibits the interactions of host factor like CPSF6 or nucleoporins with RTC/PIC complexes or, alternatively, by studying the interactions of such host factors in the absence of PF74. The important point here is that (as the authors themselves point out) their approach shows great promise for wide applications in studying host RTC/PIC-host factor interactions, but it is a significant shortcoming that such applications are not explored at all in the current study.

3) The authors should present additional data to address the effects of virus infection in the presence of APC, as the drug is used throughout HeLa imaging experiments. The authors cite a review article to indicate APC is compatible with infection, but data should be shown in their system because this step is critical to the technology. Specifically, please expand Figure 1—figure supplement 3 by two bars to show infectivity of the viruses in presence of 6 microM APC.

4) A further concern relates to the quantitative aspect of this study. For example, in the Results section (“Identification of HIV-1 RTC/PIC”), it is stated that 13 RTC/PIC objects were detected in two cells using PALM/dSTORM using TIRF imaging. This method illuminates approximately 200 nm from the glass surface attached to the bottom of the cells; but, in the same section (and elsewhere), it is stated that an average of 5 RTC/PIC (and up to 30) objects were detected using spinning disk confocal (to image the complete cellular volume). This counting incongruence (and also the possible incongruence with the reported MOI of 25, see below) brings into question the quantitative aspect of the study. Please explain or address.

5) The time course of RTCP/PIC formation depends on the sensitivity of the spinning disk imaging system. There is no mention of the minimal number of fluorescent molecules that could be detected at a single diffraction-limited spot. This is an important point that needs to be determined experimentally because it would be very easy to miss the early time points of HIV replication. The authors also need to determine the fluorescence intensity distribution associated with the EGFP signal, to understand whether they are detecting a single particle, a cluster, etc.

---

## [Author Response]

*1) The authors should investigate HIV-1 capsid uncoating in macrophages to the point where they* can *draw definitive conclusion(s). This will require imaging sufficient numbers of core particles for capsid content, and may also involve the use of other tricks to examine the status of RTC/PIC complexes (e.g., SAMHD1 depletion or Vpx treatment). These are admittedly challenging experiments, but without them the current study does not resolve the current controversy over where and when capsid uncoating occurs*.

Our previous experiments, reported in the initial submission of the manuscript, had shown that the CA staining of RTC/PIC in primary human macrophages (MDM) appeared to be more variable than observed for HeLa-based reporter cell lines. We have now performed a large number of additional experiments with primary MDM. In the revised version, we show data for a total of 135 cytoplasmic RTC/PIC and 70 nuclear PIC from a total of five independent MDM preparations from eight different donors (567 MDM imaged in total). We decided not to use SAMHD1 depletion or other tricks to enhance infection, because this may actually affect RTC/PIC composition and should thus become the topic of future functional studies. We made use of a second macrophage-tropic envelope in these studies, which was directly derived from the cerebrospinal fluid of an HIV-infected person (Env-4059). The results were identical for both macrophage-tropic Env proteins, and were thus combined in the current manuscript.

Our extended data set confirms the initial observation that cytoplasmic RTC/PIC are less often associated with detectable CA in macrophages compared to HeLa-based reporter cell lines. Only 44% of cytoplasmic RTC/PIC contained detectable CA in MDM (compared to >97% in HeLa-based reporter cells), and those cytoplasmic RTC/PIC in macrophages that did contain CA appeared to be more variable in the intensity of CA staining. Unexpectedly, however, we found that almost all nuclear PIC (68 of 70) contained strong CA staining, even in cells where we could detect cytoplasmic RTC/PIC lacking CA. This is a completely novel and unexpected result that suggests that intact capsids or capsid-like structures enter the nucleus in MDM, while most or all of CA appears to be dissociated from nuclear PICs in the reporter cells. The lower percentage of CA-positive structures in the cytoplasm of MDM may indicate that the CA-negative RTC/PIC could be dead-end structures that lose their capsid shell prior to being able to enter the nucleus. These results are now shown in the new Figure 7 (which contains some of the data from the previous MDM analysis), and quantitation is reported in the new Table 1 and [Supplementary-material SD1-data]. We believe that these additional results now allow us to draw definitive conclusions, which go far beyond the initial submission.

*2) The authors should demonstrate that their methodology* can *be used to examine RTC/PIC interactions with host factors in a meaningful way. This could be done by performing imaging experiments that demonstrate directly that PF74 inhibits the interactions of host factor like CPSF6 or nucleoporins with RTC/PIC complexes or, alternatively, by studying the interactions of such host factors in the absence of PF74. The important point here is that (as the authors themselves point out) their approach shows great promise for wide applications in studying host RTC/PIC-host factor interactions, but it is a significant shortcoming that such applications are not explored at all in the current study*.

Based on our results with PF74 at the lower concentration, we speculated that PF74 may block the CPSF6 binding interface on CA at this concentration, and may thus lead to a loss of CPSF6 association with cytoplasmic RTC/PIC. When performing immunostaining against CPSF6, we noticed that the large majority of CPSF6 is nuclear with only a very weak cytoplasmic CPSF6 signal; this had also been reported by others. Approximately 22% of cytoplasmic RTC/PIC were associated with detectable amounts of CPSF6 in our experiments for wild-type infection in the absence of PF74, and this is shown in the new Figure 6. Association of CPSF6 with RTC/PIC below the level of sensitivity cannot be ruled out for the CPSF6-negative cytoplasmic RTC/PIC in this experimental system, however. We therefore performed additional experiments in a cell line with a constitutive knock-down of TNPO3, which serves as nuclear import factor for CPSF6. In this cell line, there is significantly more cytoplasmic CPSF6 as shown in the new Figure 6—figure supplement 1. Infecting the TNPO3 knock-down cell line with HIV-1 in the absence of PF74 revealed association of CPSF6 with cytoplasmic RTC/PIC in 87% of all cases, and the presence of 2 µM PF74 during infection reduced this number to 18%. Thus, our additional results directly prove the association of CPSF6 with cytoplasmic RTC/PIC and the inhibition of this association by 2 µM PF74. Accordingly, these additional results clearly demonstrate that the new method can be used to examine RTC/PIC interaction with host factors in a meaningful way as requested by the reviewers.

*3) The authors should present additional data to address the effects of virus infection in the presence of APC, as the drug is used throughout HeLa imaging experiments. The authors cite a review article to indicate APC is compatible with infection, but data should be shown in their system because this step is critical to the technology. Specifically, please expand*
Figure 1—figure supplement 3
*by two bars to show infectivity of the viruses in presence of 6 microM APC*.

We have determined the effect of 6 µM APC on HIV-1 infection in the HeLa-based reporter cell line using either reporter expression or FACS analysis of newly produced HIV-1 CA as readout. We observed that APC treatment led to a modest reduction in infectivity by ∼50%, and this is now explicitly stated in the revised manuscript and shown in Figure 1—figure supplement 3.

*4) A further concern relates to the quantitative aspect of this study. For example, in the Results section (“Identification of HIV-1 RTC/PIC”), it is stated that 13 RTC/PIC objects were detected in two cells using PALM/dSTORM using TIRF imaging. This method illuminates approximately 200 nm from the glass surface attached to the bottom of the cells; but, in the same section (and elsewhere), it is stated that an average of 5 RTC/PIC (and up to 30) objects were detected using spinning disk confocal (to image the complete cellular volume). This counting incongruence (and also the possible incongruence with the reported MOI of 25, see below) brings into question the quantitative aspect of the study. Please explain or address*.

The number of RTC/PIC per individual cell varied over a wide range in all experiments that we have performed. This may not be surprising given the stochasticity of the infection process. Accordingly, we observed cells that carried 0 RTC/PIC and cells that carried up to over 40 RTC/PIC in the same infection experiment (at 4.5 hr p.i.). To clarify this issue, we are now providing a bar graph showing the distribution of numbers of RTC/PIC per cell in the experiments used in this study (Figure 1—figure supplement 7). The same distribution also occurred in the PALM/dSTORM experiments and we of course used cells, which happened to display a larger number of RTC/PIC in the TIRF field. Many other cells exhibited no RTC/PIC in the TIRF field in the same analysis.

*5) The time course of RTCP/PIC formation depends on the sensitivity of the spinning disk imaging system. There is no mention of the minimal number of fluorescent molecules that could be detected at a single diffraction-limited spot. This is an important point that needs to be determined experimentally because it would be very easy to miss the early time points of HIV replication. The authors also need to determine the fluorescence intensity distribution associated with the EGFP signal, to understand whether they are detecting a single particle, a cluster, etc*.

If we understand correctly, this comment addresses two different issues: (i) sensitivity of the experimental setup, i.e. what number of fluorophores can be determined and (ii) are we looking at single particles or clusters.

i) To determine the sensitivity of our experimental setup, we performed a dilution experiment with the dye. To this end, we sequentially diluted Alexa Fluor 647 azide (the dye used for EdU click labelling) from 3.5 mM to 3.5 nM and imaged it in the SDCM under the same conditions of detection as used for RTC/PIC. These experiments indicated that the sensitivity of our system is almost single molecule (i.e., 1-2 molecules could be detected). The results of this experiment are shown in the new Figure 1—figure supplement 6 in the revised version. Of course, sensitivity for RT products will not be as high since it depends in addition on the number of T-residues in the respective nucleic acid, the relative incorporation of T vs EdU under the experimental conditions and the efficiency of click labelling. We therefore performed additional experiments with lentiviral vectors with lengths of 5.4 or 3.7 kb, respectively (∼2.600 or ∼1.800 T residues for double-stranded DNA), which were also detectable in our experimental system (new Figure 1—figure supplement 6).

ii) Determining the fluorescence intensity distribution of the EGFP signal would not appear to be optimal to distinguish between clusters or single particles. Vpr.IN.eGFP incorporation into individual particles varies over a rather wide range as we observed when analyzing the fluorescence of individual particles adhered to cover slips. Furthermore, it cannot be predicted how many IN.EGFP molecules will stay with the cytoplasmic RTC/PIC after entry. To address the issue of single particles vs clusters, we determined the diameter of the 16 RTC/PIC that were observed by three-color PALM/dSTORM microscopy. The results show a diameter of ∼155 nm for CA and ∼133 nm for IN.mEos with a modest variation and no clear outliers (new Figure 4—figure supplement 1). Accordingly, we conclude that RTC/PIC detected in our system represent individual particles and not clusters.